# The Impact of Age on Mortality in Chronic Haemodialysis Population with COVID-19

**DOI:** 10.3390/jcm10143022

**Published:** 2021-07-07

**Authors:** Ander Vergara, Mireia Molina-Van den Bosch, Néstor Toapanta, Andrés Villegas, Luis Sánchez-Cámara, Patricia de Sequera, Joaquín Manrique, Amir Shabaka, Inés Aragoncillo, María Carmen Ruiz, Silvia Benito, Emilio Sánchez, María José Soler

**Affiliations:** 1Nephrology Department, Vall d’Hebrón University Hospital, 08035 Barcelona, Spain; avergara@vhebron.net (A.V.); ntoapanta@vhebron.net (N.T.); 2Nephrology Research Group, Vall d’Hebrón Research Institute, REDinREN, 08035 Barcelona, Spain; mireiamoli@gmail.com; 3Nephrology Department, Germans Trias i Pujol University Hospital, 08916 Badalona, Spain; anvillegas.germanstrias@gencat.cat; 4Nephrology Department, Fundación para la Investigación Biomédica Gregorio Marañón, 28007 Madrid, Spain; sanchezcamaramd@gmail.com (L.S.-C.); inesiglia@hotmail.com (I.A.); 5Nephrology Department, Infanta Leonor University Hospital, 28031 Madrid, Spain; psequerao@senefro.org; 6Nephrology Department, Complejo Hospitalario de Navarra, 31008 Pamplona, Spain; joaquin.manrique.escola@navarra.es; 7Nephrology Department, Navarra Institute for Health Research, IdiSNA, 31008 Pamplona, Spain; 8Nephrology Department, Fundación Alcorcón University Hospital, 28922 Madrid, Spain; amirshabaka@hotmail.com; 9Nephrology Department, Virgen de las Nieves University Hospital, 18014 Granada, Spain; mamenruizfuentes@telefonica.net; 10Nephrology Department, Fundació Puigvert, IIB Sant Pau, 08025 Barcelona, Spain; sbenito@fundacio-puigvert.es; 11Nephrology Department, Cabueñes University Hospital, 33394 Gijón, Spain; jesastur@gmail.com

**Keywords:** COVID-19, elderly, haemodialysis, mortality, SARS-CoV-2

## Abstract

Age and chronic kidney disease have been described as mortality risk factors for coronavirus disease 2019 (COVID-19). Currently, an important percentage of patients in haemodialysis are elderly. Herein, we investigated the impact of age on mortality among haemodialysis patients with COVID-19. Data was obtained from the Spanish COVID-19 chronic kidney disease (CKD) Working Group Registry. From 18 March 2020 to 27 August 2020, 930 patients on haemodialysis affected by COVID-19 were included in the Registry. A total of 254 patients were under 65 years old and 676 were 65 years or older (elderly group). Mortality was 25.1% higher (95% CI: 22.2–28.0%) in the elderly as compared to the non-elderly group. Death from COVID-19 was increased 6.2-fold in haemodialysis patients as compared to the mortality in the general population in a similar time frame. In the multivariate Cox regression analysis, age (hazard ratio (HR) 1.59, 95% CI: 1.31–1.93), dyspnea at presentation (HR 1.51, 95% CI: 1.11–2.04), pneumonia (HR 1.74, 95% CI: 1.10–2.73) and admission to hospital (HR 4.00, 95% CI: 1.83–8.70) were identified as independent mortality risk factors in the elderly haemodialysis population. Treatment with glucocorticoids reduced the risk of death (HR 0.68, 95% CI: 0.48–0.96). In conclusion, mortality is dramatically increased in elderly haemodialysis patients with COVID-19. Our results suggest that this high risk population should be prioritized in terms of protection and vaccination.

## 1. Introduction

In December 2019, a novel coronavirus emerged as the cause of a cluster of pneumonia cases in Wuhan (China). The virus was designated as severe acute respiratory syndrome coronavirus 2 (SARS-CoV-2) and has rapidly spread worldwide. As of 13 June 2021, more than 175 million cases of coronavirus disease 2019 (COVID-19) and more than 3.8 million deaths due to the infection have been reported to the World Health Organization (WHO) [1], demonstrating that we are facing the biggest health crisis of the new century.

To date, several risk factors have been attributed to worse COVID-19 outcomes like older age, male sex and various comorbidities such as diabetes, obesity or hypertension [2]. Amongst these, age is an important and independent risk factor for mortality in COVID-19 patients [2,3]. Chronic kidney disease (CKD) has also been described as a risk factor for severity of COVID-19 and related to a greater risk of death [4,5]. Williamson et al. performed a study from OpenSAFELY health analytics platform including data from more than 17 million people in the UK to identify risk factors for COVID-19 mortality. Their results also highlighted the importance of CKD as a risk factor for COVID-19 mortality [6]. According to the 2019 report of the Spanish Registry of Chronic Kidney Disease Patients, 70% of patients in KRT (Kidney Replacement Therapy) aged 75 or older are under chronic haemodialysis program [7]. Similarly, 42% of patients aged between 65 and 74 years that require KRT are on haemodialysis [7]. The mean age of patients in haemodialysis in Spain is 68 years old [8]. In contrast, kidney transplantation is the leading KRT in young adults (<65 years old) with end-stage kidney disease (ESKD).

Considering that most of the patients in haemodialysis are elderly patients and that both age and CKD have been identified as mortality risk factors, we sought to analyze mortality in elderly haemodialysis patients (≥65 years) with COVID-19, and evaluate the risk factors for mortality in this population. In addition, we also compared the differences in mortality between the elderly haemodialysis population and the elderly general population.

## 2. Materials and Methods

### 2.1. Data Source

The present study reports data obtained from the Spanish COVID-19 CKD Working Group Registry. The Registry is a Spanish multi-centre database that was created on 18 March 2020. The Spanish Society of Nephrology and a special expert committee created for this task are responsible for keeping and authorizing the use of the Registry data. In the database, information of patients on KRT (haemodialysis, peritoneal dialysis and kidney transplantation) with confirmed COVID-19 is recorded. Medical doctors, responsible for the KRT COVID-19 positive patients’ assistance, enter the data by filling the boxes in a web form. Only patients with a confirmed COVID-19 by a positive result on a real-time polymerase chain reaction (PCR) assay of nasal or pharyngeal swab specimens can be included. Patients in KRT were eligible when they had a previously diagnosed end-stage kidney disease (ESKD) or when they remained in KRT for 3 months or more. Both outpatients and hospitalized patients were included in the Registry. Only a limited number of variables related to baseline characteristics (clinical manifestations, outcome, and specific treatments) were included in the web form to ensure the proper completion of the Registry [9].

The Spanish COVID-19 CKD Working Group Registry was initially approved by The Ethics Committee of the Principality of Asturias (Project number: 2020.233). In addition, the analysis of the specific data of patients in haemodialysis and the current study protocol were approved by the Vall d’Hebron Ethics and Medical Research Committee (Project number: PR(AG)474/2020).

Data from the general population infected by COVID-19 was obtained from the Spanish National Epidemiological Surveillance Network. The agency is linked to the Health Ministry, and has been publishing a weekly report since the first COVID-19 diagnosis in Spain. These reports are public and contain cumulative cases and deaths. Data were collected from 31 January to 27 August 2020, which includes a similar period to the analyzed in our Registry [10,11].

### 2.2. Data Collection

Data analysis for the current study has been restricted to haemodialysis patients. The collection period included from 18 March to 27 August 2020. All-cause mortality was the primary COVID-19 outcome evaluated in both univariate and multivariate analyses. The follow-up period covers the time elapsed in days until the end of the follow-up recorded in each patient, or the last data collection date on 27 August 2020. No specific criteria was used to define the end of follow-up. In most cases, patient discharge was considered the end of follow-up. Information collected included baseline characteristics (age, sex, or primary kidney disease), COVID-19 related characteristics (clinical manifestations, pneumonia, or the need for hospitalization) and specific treatments received for the infection. Patients with a positive polymerase chain reaction (PCR) for SARS-CoV-2 who did not develop any symptoms during follow-up were considered asymptomatic. A radiological image obtained by different techniques (chest X-ray, computed tomography or magnetic nuclear imaging), was used to diagnose pneumonia. Mechanical ventilation variable included both invasive and non-invasive ventilation. Other variables of interest were haemodialysis location (haemodialysis performed in the hospital, in an outpatient dialysis centre or at home), the prior use of renin-angiotensin system blockers or non-steroidal anti-inflammatory drugs. The severity of COVID-19 was assessed by signs and symptoms of the disease, namely dyspnea, pneumonia or the need of mechanical ventilation.

### 2.3. Statistical Analysis

Data management and statistical analysis were performed with Stata software 15.1 version (College Station, TX, USA). Continuous quantitative variables are summarised as mean with standard deviation (SD) or median and interquartile range (IQR) when the variable shows a non-normal distribution. Qualitative variables are displayed as frequencies and within-group percentages. Baseline characteristics between elderly and non-elderly populations were evaluated with Pearson’s chi-squared test (χ^2^) for qualitative data and Student’s t-test for quantitative data. Confidence intervals for proportions were calculated with the Wilson method. Differences between the haemodialysis and the general population were evaluated with Pearson’s chi-squared test (χ^2^). The capacity of age to predict mortality was assessed with receiver operating characteristic (ROC) curves. The cut-off age with the best sensitivity and specificity was established with these curves finding the nearest (0, 1) value. Univariate analysis was performed using Cox proportional hazard regression analysis. Hospitalization was considered an essential factor in disease prognosis, as hospitalized patients usually develop more severe disease. Moreover, patients admitted to the hospital were more likely to receive COVID-19 specific treatments. In order to avoid this bias, a univariate analysis of treatments was performed only in hospitalized elderly haemodialysis patients (see Appendix A). Hospitalization was also included in the multivariate models to be adjusted as a possible confounding factor. Multivariate analysis was performed using Cox proportional hazards model. All models included age, sex and hospitalization, as the three factors were statistically associated with mortality in univariate analysis and had been previously identified as mortality risk factors. A first model included only the clinical manifestations statistically associated with higher mortality and adjusted for age, sex, and hospitalization in order to identify the signs or symptoms of a worse prognosis. The second and third model were performed to study variables of interest after adjusting for patients’ disease severity. The second model included primary kidney disease adjusted to age, sex, hospitalization, and clinical manifestations. The third model included specific COVID-19 treatments adjusted for age, sex, hospitalization, and clinical manifestations. A *p*-value <0.05 indicated statistical significance.

## 3. Results

From 18 March to 27 August, 1812 patients were included in the Spanish COVID-19 CKD Working Group registry. Of those, 1169 were under chronic haemodialysis. Six of these patients performed haemodialysis at home. The remaining 1163 patients were included in an in-centre haemodialysis program (hospitals and outpatient dialysis centres). Four patients without birth dates and 86 patients with less than 3 months on hemodialysis were excluded from the study. Of the 1079 haemodialysis patients included in the study, 297 (27.5%) were younger than 65 years of age at the date of COVID-19 diagnosis, and 782 (72.5%) patients were 65 years or older, the latter was defined as the elderly group. Within the elderly group, 269 were between 65 and 74 years, and 513 were ≥75 years. Among the elderly group, 88 patients did not have the infection outcome (all-cause mortality) recorded, and for 18 patients, the follow-up could not be estimated. Thus, these patients were excluded from the analysis. A total of 676 elderly haemodialysis patients were included in the univariate and multivariate analyses (Figure 1).

### 3.1. Differences between Elderly and Non-Elderly Haemodialysis Populations

Baseline characteristics of elderly and non-elderly patients in haemodialysis program are summarised in Table 1. There were no differences in sex between both groups. Nephrosclerosis as a cause of end-stage kidney disease was 10.8% (95% CI: 6.5–15.0) higher in elderly patients, while glomerular diseases were a more common cause of renal disease in the younger population. Elderly patients depicted a more severe COVID-19. They had an increased risk of developing dyspnea (HR 2.61, 95% CI: 2.04–3.35) or pneumonia (HR 5.10, 95% CI: 3.52–7.38) at presentation. Lymphopenia was also higher in this population (71.2% vs. 63.6%). Hospital admission was more frequent in elderly patients (HR 9.03, CI 95%: 5.17–15.76) compared to the younger haemodialysis population. However, they did not receive more invasive or intensive care therapies. Aged haemodialysis patients were less frequently admitted into intensive care units (ICU) or received mechanical ventilation therapies less often in comparison with non-elderly haemodialysis population. In fact, ICU admission was denied due to comorbidities or resource allocation in 12.4% of the patients in the elderly group as compared to 7.9% in the younger group.

### 3.2. Mortality in Haemodialysis Patients

Overall mortality in haemodialysis patients infected with COVID-19 was 30.0%. All-cause mortality was 19.6% higher (95% CI: 13.8–25.4%, *p* < 0.001) in the elderly haemodialysis patients compared to the non-elderly population. Thus, the risk of death was 2.57-fold (95% CI: 1.84–3.59) higher in the aged patients. Within the elderly population, patients aged 75 or more had a higher risk of dying (HR 1.28, 95% CI: 0.97–1.69) than those aged between 65 and 74. These differences observed in the elderly group were not statistically significant (*p* = 0.078). Nevertheless, there is a trend of increasing mortality with age, evidenced in the survival analysis (Figure 2). To assess the impact of age on predicting mortality in haemodialysis population we plotted ROC curves using logistic regression analysis (see Appendix A). Although statistically significant, age alone was a weak predictor of mortality with an AUC of 0.637 (95% CI: 0.605–0.668). The age cut-off value was estimated using the same ROC curves. 74.8 years was the nearest (0, 1) value with a sensitivity of 60.9% and a specificity of 57.0%. Therefore, in our Registry, patients that died had a 60.9% probability of being older than 74.8 years, whereas the survivors had a 57.0% probability of being younger than 74.8 years. An increase in sensitivity to 90% modified the cut-off to 61.4 years. Thus, in our dataset, the probability that non-survivors were older than 61.4 years was 90%, showing that mortality was more frequent in elderly patients.

### 3.3. Mortality Compared to the General Population

Mortality in haemodialysis patients was 25.1% higher (95% CI: 22.2–28.0%, *p* < 0.001) as compared to mortality in the general population during a similar time frame [10,11]. Available baseline characteristics between both populations are shown in Table 2. Haemodialysis patients were older and male sex was more frequent. Of haemodialysis patients with COVID-19 recorded in our Registry, 30.0% died in contrast to 4.9% in the general population. This difference is observed in every age range (Figure 3 and Appendix A) and represents an increase of 6.2-fold (Prevalence ratio (PR) 6.17, 95% CI: 5.59–6.81) in mortality. When estimating the risk, patients on haemodialysis had eight times more risk of dying from COVID-19 than the general population (OR 8.38, 95% CI: 7.28–9.65). Hospitalization was also 50.9% higher (95% CI: 48.1–53.7%, *p* < 0.001) and haemodialysis patients had a nine times higher risk of being admitted to the hospital (OR 9.11, 95% CI: 7.88–10.53). In the general population, 23.1% (95% CI: 23.0–23.2) of patients with COVID-19 were hospitalized compared to 74.0% (95% CI: 71.2–76.8) of haemodialysis patients.

### 3.4. Mortality Risk Factors in the Elderly Haemodialysis Population

In the elderly group, the follow-up time was 20 (Interquartile range (IQR): 14–36) days and 7 (IQR: 3–13) days for the survivors and non-survivors, respectively (Table 3). Advanced age was associated with increased mortality. A 10-year age increase was related to a 1.4-fold greater risk of dying. In addition, male sex was also associated with a higher mortality. Within the causes of primary kidney disease, an increased risk of death was observed in patients with diabetic kidney disease. On the contrary, CKD secondary to interstitial nephropathy was associated with a lower risk of mortality in the univariate Cox regression analysis (Table 3). When compared with at home or in an outpatient dialysis service, in-hospital haemodialysis showed lower mortality (HR 0.73, 95% CI: 0.55–0.98). Treatment with angiotensin receptor blockers, angiotensin-converting enzyme inhibitors or non-steroidal anti-inflammatory drugs prior to infection had no impact on the outcome.

Regarding the clinical manifestations, elderly patients that died had a more severe presentation of the disease. Fever, respiratory symptoms, such as cough, rhinorrhea or sputum production, as well as dyspnea, were associated with a worse prognosis (Table 3). Dyspnea at presentation was the worst symptom with a 2.3-fold increase in mortality. Conversely, patients that were asymptomatic at diagnosis showed a lower risk of death. Gastrointestinal symptoms, namely nausea, vomiting or diarrhea, had no impact on the patient outcome. COVID-19 pneumonia at diagnosis worsened the disease prognosis, and it is a significant factor that increases the risk of death almost 4-fold. Lymphopenia was also more frequent in patients with severe disease and was associated with higher mortality in the univariate analysis.

Admission to hospital and the need for intensive care or mechanical ventilation were other critical factors that worsened the prognosis of the disease. Elderly patients on haemodialysis admitted to hospital were more likely to die (HR 7.38, 95% CI: 4.12–13.20). Only 66.6% of the survivors were hospitalized as compared to 93.7% of non-survivors, which is a 26.9% increase in hospitalization percentage (95% CI: 21.8–32.0%, *p* < 0.001) in the non-survivors (Table 3). Surprisingly, once admitted, longer hospital stays were associated with a better prognosis (hazard ratio for five days of hospitalization of 0.59, 95% CI: 0.52–0.67). The latter finding might be in part explained because COVID-19 is an acute illness with high mortality in the first week after admission to hospital [12,13]. Patients who survived the first week of hospitalization had a greater probability of being discharged, although they had a longer hospitalization stay. Admission to ICU or the need for mechanical ventilation was related to more severe disease and higher mortality in elderly haemodialysis patients. Mechanical ventilation included both patients that received invasive and non-invasive techniques. Although elderly patients were rarely admitted to the ICU (1.3%), a significant number (7.4%) of older patients received mechanical ventilation therapies during hospitalization, the vast majority of which were probably non-invasive (Table 1).

In the adjusted Cox models, age, dyspnea at presentation, pneumonia or the need for hospitalization were identified as risk factors for mortality in the elderly haemodialysis population with COVID-19. These four factors prevailed as independent risk factors for mortality in all the three regression models performed in the present study (Table 4). Admission to hospital increased the risk of death four- to six-fold, whereas dyspnea at presentation or the diagnosis of pneumonia increased this risk 1.5-2-fold. After adjusting for age, hospitalization and clinical severity of the disease, we performed another two Cox regression models to identify whether primary kidney disease or specific COVID-19 treatments had an impact on mortality (Table 4). The regression model including the primary kidney disease showed that only CKD secondary to systemic diseases was independently associated with mortality. Nevertheless, it is difficult to draw any conclusions from this finding, as only 14 elderly patients with systemic disease as a cause of ESKD were included. Surprisingly, diabetic kidney disease, which was related to a higher risk of mortality in the univariate analysis, was not identified as a risk factor in the multivariate analysis. The model including specific COVID-19 therapies, demonstrated that glucocorticoid treatment was independently associated with a lower risk of mortality.

### 3.5. Treatment of COVID-19 in the Elderly Haemodialysis Population

We studied the specific treatments for COVID-19 administered to the elderly population on haemodialysis during the pandemic. Treatments included were lopinavir-ritonavir (LPV/r), hydroxychloroquine (HCQ), interferon beta, tocilizumab, glucocorticoids and azithromycin. Without adjusting the elderly population for hospitalization, treatment with LPV/r, HCQ or interferon beta were associated with increased mortality. Non-hospitalized patients usually had an asymptomatic or mild disease, without evidence of pneumonia, and did not receive any specific treatment. Therefore, analyzing the latter group together with patients with a more severe COVID-19 could explain this finding in the unadjusted analysis. After adjusting for hospitalization, no particular treatment showed an impact in reducing mortality (see Appendix A). In the Cox multivariate analysis (Table 4), only glucocorticoids were identified as a protective treatment for lowering mortality in elderly haemodialysis patients with COVID-19 (hazard ratio 0.68, 95% CI: 0.48–0.96; *p* = 0.027).

## 4. Discussion

The present study focuses on mortality in elderly haemodyalisis (≥65 years-old) patients affected from COVID-19 and recorded in the Spanish COVID-19 CKD Working Group Registry. We demonstrated that the risk of dying in this population was almost three times higher (OR 2.93, 95% CI: 2.02–4.24) when compared to non-elderly haemodialysis patients (<65 years-old). Age, dyspnea at presentation, pneumonia or the need for hospitalization were identified as independent risk factors for mortality in the elderly haemodialysis patients with COVID-19. Interestingly, glucocorticoid therapy was identified as a protective factor for mortality in this population.

Mortality in the haemodialysis population with COVID-19 is dramatically high. Thus, when comparing the crude mortality in haemodialysis patients in our Registry with the mortality in the general population during a similar time frame (from 31 January to 27 August 2020) from the Spanish National Epidemiological Surveillance Network, a 6.2-fold mortality increase was identified in the haemodialysis population [10,11]. Crude mortality for the haemodialysis patients included in our database was 30.0%, whereas it was 4.9% in the Spanish population. These findings could be, in part, explained by the fact that patients in our Registry were older, with a higher proportion of males (64.1%) and probably a more significant number of comorbidities than the general population. Nevertheless, after adjusting mortality to age, an increased risk of death was observed in all ranges for the haemodialysis population, being extremely high (>30%) in hemodialysis patients older than 70 years (Figure 3). Similar increases in the risk of death attributable to haemodialysis or CKD were previously described, identifying a 2-fold mortality increase in haemodialysis patients compared to the general population [14,15]. United Kingdom population data analysis obtained from the OpenSAFELY health platform also demonstrated that CKD is a significant risk factor for COVID-19 mortality [6,16]. Cardiovascular risk factors, morbidity usually related to ESKD and age may in part explain the increase in mortality observed in the haemodialysis population. Other pathological conditions common in CKD patients, namely, diabetes, hypertension, obesity or heart failure, have already been identified as risk factors for severe COVID-19 and death in the general population [2,3,6]. Of note that the high mortality in elderly hemodialysis patients may also be ascribed to the fact that these patients did not receive more invasive or intensive care therapies in comparison with the non-elderly population on hemodialysis. Moreover, elderly patients possibly showed a more severe immunosuppression state as lymphopenia was more frequent in this group and has been related to a greater risk of mortality in previous series [17].

Male sex has been identified as a risk factor for mortality in the general population with COVID-19 [2,6]. In the univariate analysis of our study, male sex was also associated with death in the elderly haemodialysis population. However, this significant difference was lost in the multivariate analysis. The effect of gender on the severity and prognosis of COVID-19 in KRT patients is controversial. While some studies showed increased mortality in males [15], others could not find any difference between sexes [13]. If confirmed, the worse survival rate in males with COVID-19 may be in part ascribed to the increase in circulating angiotensin converting enzyme 2 (ACE2) levels [18], which could facilitate the systemic spread of SARS-CoV-2. Whether sex has any impact on mortality or the implications of ACE2 should be assessed in further studies.

Elderly haemodialysis patients developed a more severe COVID-19 as compared to young haemodialysis patients. Age was identified as a significant predictor of mortality with an AUC of 0.637 (see Appendix A). An age cut-off value of 74.8 years was established with an efficiency for correctly classifying survivors and non-survivors of almost 60%. Moreover, in the aged population, dyspnea or pneumonia at presentation were more frequent, and both clinical manifestations were independent risk factors for mortality in all the multivariate models performed in this study. Thus, haemodialysis patients aged 65 or older that showed fever, cough, sputum production or dyspnea at presentation had a higher risk of mortality. In a recent study performed in kidney transplant recipients, pneumonia was also associated with a worse prognosis, but gastrointestinal symptoms alone were associated with a decreased risk of death [19]. However, in the present study, gastrointestinal symptoms at diagnosis, namely diarrhea, nausea or vomiting, had no impact on prognosis. As expected, being asymptomatic at diagnosis was related to a mild disease, and it implied a better prognosis. In our study, elderly patients were more frequently admitted into the hospital, which was a powerful predictor of mortality. Hospital admission increased the risk of death in the aged population four to six times in haemodialysis patients with COVID-19 compared to non-admitted patients. In line with these findings, studies in patients with ESKD with COVID-19 have also identified age, higher comorbidly indexes, frailty or pneumonia as independent risk factors for mortality [9,13,14].

Among specific treatments for SARS-CoV-2 infection, only glucocorticoids were identified as a protective factor for mortality in elderly haemodialysis patients. These results suggest that glucocorticoids may become a good therapeutic option for this specific population. Similar results have been reported in the general population, demonstrating that different regimens of glucocorticoids, especially those using dexamethasone, are able to reduce mortality [20,21,22]. Sterne et al. recently published a meta-analysis that pooled data from seven different randomized clinical trials. The results revealed that treatment with glucocorticoids significantly decreased the risk of death. When studying the effect of the different types of glucocorticoids included in the analysis, only dexamethasone demonstrated a significant protection. However, dexamethasone was the glucocorticoid type with the largest sample size within the meta-analysis [20]. In concordance with our study, other authors also highlighted the potential benefits of corticosteroids in the treatment of moderate–severe COVID-19 patients [23]. In our database, the employed glucocorticoid or duration of treatment was not recorded and probably includes a wide range of types and doses. For this reason, it is not possible to recommend a concrete regimen and further studies are needed to confirm this finding.

### Strengths and Limitations

This study reports the data of a representative sample of elderly haemodialysis patients with COVID-19 from the Spanish COVID-19 CKD Working Group Registry. It includes 930 patients on haemodialysis diagnosed throughout the Spanish territory. The sample represents 3.6% of all haemodialysis patients in Spain, considering the total number of haemodialysis patients included in the 2019 report of the Spanish Registry of Chronic Kidney Disease Patients [7]. Within the 930 patients included, 676 reflect the characteristics and risk factors for mortality of the main haemodialysis population in Spain and Europe: the elderly patients [7,24]. Such a number of patients allow us to develop more solid conclusions when compared to smaller single-centre studies. Furthermore, the comparison of the mortality on the haemodialysis population with that of the general population for a similar period and adjusted to age, evidences the dramatic consequences of COVID-19 on haemodialysis. However, and despite the considerable sample size, the retrospective nature of the Registry cannot be overlooked. Data of some patients was missing, and because of the design of the Registry, other important baseline characteristics or comorbidities were not studied nor analyzed as possible risk factors for mortality. Moreover, the long-term follow-up data was probably incomplete in many patients which may help to explain the increased number of censored patients and the flattening of the survival curves after 50 days.

It is of note that only all-cause mortality was recorded in our Registry. Despite the fact that almost all patients died in the first 50 days after infection diagnosis, we cannot accurately assure if death in elderly patients occurred secondary to COVID-19 or due to exacerbation of previous conditions. However, as in most patients the main diagnosis was COVID-19, one assumes that the mortality should be mainly ascribed to COVID-19 and its complications. In addition, because of the disease’s nature, some asymptomatic patients infected by SARS-CoV-2 were probably not diagnosed and included in the Registry.

## 5. Conclusions

Elderly patients on haemodialysis with COVID-19 have higher mortality rates than younger haemodialysis patients or the elderly general population. Age, dyspnea at presentation, pneumonia or hospitalization have been identified as risk factors for worse prognosis in elderly haemodialysis patients with COVID-19. In addition, the present study suggests that glucocorticoids may be a therapeutic option for these patients. These results help us to understand the impact of COVID-19 on elderly hemodialysis patients and suggest that a special policy and management should be applied to avoid COVID-19 in this high-risk population. In front of COVID-19, and maybe other pandemies to come, older hemodialysis patients should be prioritized in terms of decreasing public transportation, implementation of hygienic/protective strategies, and vaccination when available.

## Figures and Tables

**Figure 1 jcm-10-03022-f001:**
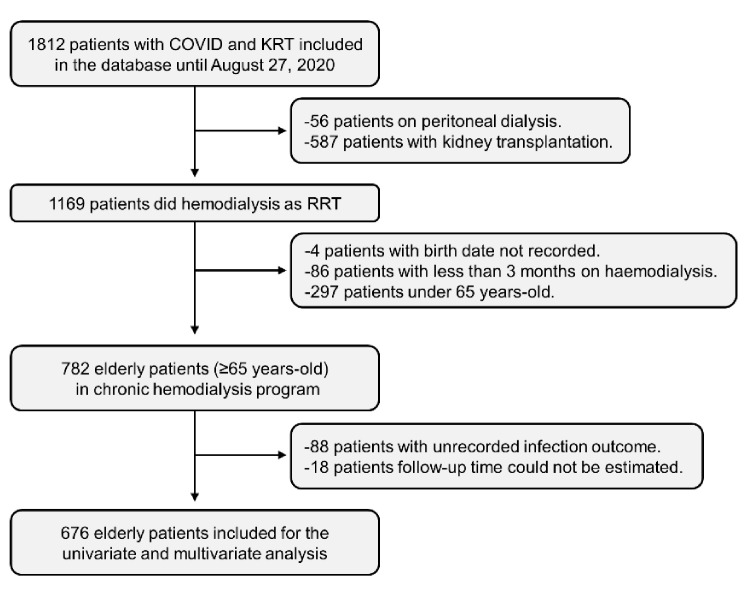
Study flow diagram. KRT: Kidney Replacement Therapy.

**Figure 2 jcm-10-03022-f002:**
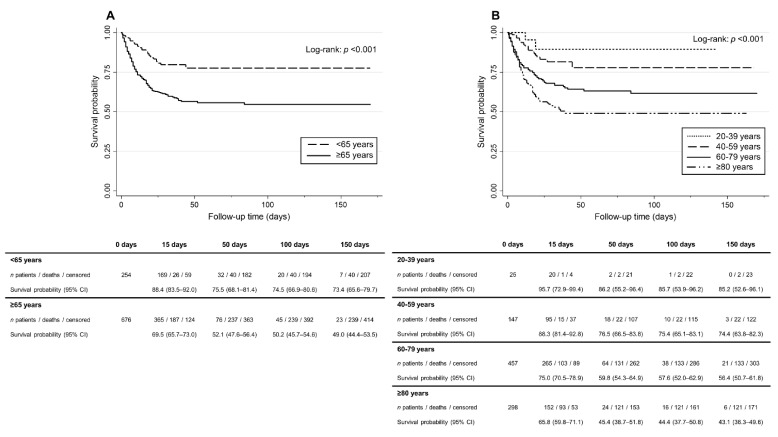
Kaplan–Meier survival curves for all-cause mortality in the haemodialysis population: distribution by age. (**A**) Survival differences between elderly (≥65 years-old) and non-elderly (<65 years-old) population; (**B**) Survival differences within haemodialysis population by age range. HD: Haemodialysis.

**Figure 3 jcm-10-03022-f003:**
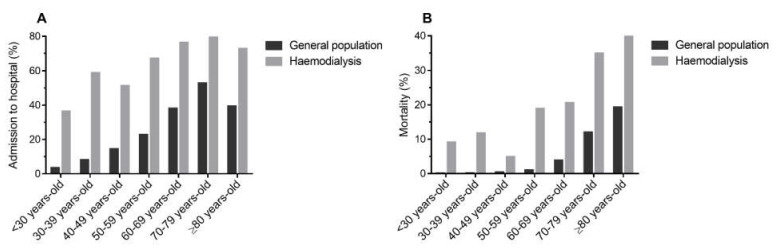
Hospitalization and mortality in Spanish general population and haemodialysis patients with COVID-19. (**A**) Percentage of patients admitted to hospital by age range in the general population and haemodialysis patients. (**B**) Mortality by age range in the general population and haemodialysis patients. General population data was obtained from the Spanish National Epidemiological Surveillance Network between 31 January 2020 and 27 August 2020 [10,11]. Haemodialysis data obtained from the Spanish COVID-19 Working Group database.

**Table 1 jcm-10-03022-t001:** Demographic and clinical features of elderly and non-elderly patients on haemodialysis with known outcome and follow-up time. Values expressed in mean ± SD or *n* (%). ICU: Intensive Care Unit. NA: Not Assessed.

Variables	<65 Years-Old (*n* = 254)	≥65 Years-Old (*n* = 676)	*p*-Value
Mean follow-up time (days)	20 (IQR: 13–30)	16 (IQR: 8–28)	NA
Age	53.5 ± 10.7	78.3 ± 7.2	<0.001
Male sex	165 (65.0%)	431 (63.8%)	0.733
Primary kidney disease:			
Diabetic kidney disease	69 (27.2%)	188 (27.8%)	0.582
Primary glomerular disease	50 (19.7%)	68 (10.1%)	<0.001
Interstitial nephropathy	15 (5.9%)	40 (5.9%)	0.643
Polycystic kidney disease	13 (5.1%)	22 (3.3%)	0.294
Nephrosclerosis	18 (7.1%)	121 (17.9%)	<0.001
Systemic disease	8 (3.2%)	14 (2.1%)	0.430
Others or unknown	81 (31.8%)	223 (32.9%)	NA
Time on hemodialysis (years)	2.4 (IQR: 1.1–5.3)	3.1 (IQR: 1.5–5.8)	0.335
Chronic haemodialysis performed in a hospital	120 (47.2%)	287 (42.5%)	0.389
Clinical presentation:			
Asymptomatic	44 (17.3%)	90 (13.3%)	0.087
Fever	163 (64.2%)	431 (63.8%)	0.912
Respiratory symptoms ^a^	138 (54.3%)	373 (55.2%)	0.933
Dyspnea	72 (28.4%)	257 (38.0%)	0.022
Gastrointestinal symptoms ^b^	42 (16.5%)	97 (14.4%)	0.641
Pneumonia	131 (51.6%)	435 (64.4%)	<0.001
Lymphopenia	160 (63.0%)	481 (71.2%)	0.002
Hospitalization	166 (65.4%)	515 (76.2%)	0.002
Admission to ICU	25 (9.8%)	9 (1.3%)	<0.001
Mechanical ventilation	27 (10.6%)	50 (7.4%)	0.177
Mortality	40 (15.8%)	239 (35.4%)	<0.001

^a^ Included cough, rhinorrhea and sputum production. ^b^ Included nausea, vomiting and diarrhea.

**Table 2 jcm-10-03022-t002:** Characteristics between haemodialysis population and Spanish general population affected by COVID-19. Values expressed in mean ± SD or *n* (%). ICU: Intensive Care Unit. General population data was obtained from the Spanish National Epidemiological Surveillance Network between 31 January 2020 and 27 August 2020 [10,11]. Haemodialysis data obtained from the Spanish COVID-19 Working Group database.

Variables	Haemodialysis Population (*n* = 930)	General Population (*n* = 438,469)	*p*-Value
Age	71.6 ± 13.8	50.8 ± 23.5	<0.001
Male sex	596 (64.1%)	201,018 (45.8%)	<0.001
Hospitalization	681 (74.0%)	101,277 (23.1%)	<0.001
Admission to ICU	34 (5.2%)	8329 (1.9%)	<0.001
Mortality	279 (30.0%)	21,325 (4.9%)	<0.001

**Table 3 jcm-10-03022-t003:** Univariate Cox regression for mortality risk factors in elderly patients on haemodialysis with COVID-19. Values expressed in median and interquartile range (IQR), mean ± SD or *n* (%). HR: Hazard ratio and 95% confidence interval for the ratio. KRT: Kidney Replacement Therapy. ICU: Intensive Care Unit. NA: Not Assessed.

Variables	Total (*n* = 676)	Survivors (*n* = 437)	Non-Survivors (*n* = 239)	HR (95% CI)	*p*-Value
Mean follow-up time (days)	16 (IQR: 8–28)	20 (IQR: 14–36)	7 (IQR: 3–13)	NA	NA
Age	78.3 ± 7.2	77.7 ± 7.0	79.5 ± 7.4	1.35 (1.13–1.62) ^c^	0.001
Male sex	431 (63.8%)	265 (60.6%)	166 (69.5%)	1.36 (1.03–1.79)	0.029
Primary kidney disease:					
Diabetic kidney disease	188 (27.8%)	117 (26.8%)	71 (29.7%)	1.44 (1.07–1.94)	0.016
Primary glomerular disease	68 (10.1%)	48 (11.0%)	20 (8.4%)	0.87 (0.55–1.39)	0.566
Interstitial nephropathy	40 (5.9%)	34 (7.8%)	6 (2.5%)	0.38 (0.17–0.86)	0.020
Polycystic kidney disease	22 (3.3%)	15 (3.4%)	7 (2.9%)	0.95 (0.44–2.01)	0.884
Nephrosclerosis	121 (17.9%)	84 (19.2%)	37 (15.5%)	0.94 (0.66–1.36)	0.754
Systemic disease	14 (2.1%)	7 (1.6%)	7 (2.9%)	1.84 (0.86–3.92)	0.114
Others or unknown	223 (32.9%)	132 (30.2%)	91 (38.1%)	NA	NA
Time on KRT (years)	3.1 (IQR: 1.5–5.8)	3.1 (IQR: 1.4–5.6)	3.1 (IQR: 1.6–6.0)	1.00 (0.99–1.02) ^d^	0.838
Chronic haemodialysis performed in a hospital	287 (42.5%)	204 (46.7%)	83 (34.7%)	0.73 (0.55–0.98)	0.034
Received treatments prior to infection:					
Angiotensin converting enzyme inhibitors (ACEi)	58 (8.6%)	36 (8.2%)	22 (9.2%)	1.16 (0.75–1.80)	0.501
Angiotensin receptor blockers (ARB)	84 (12.4%)	62 (14.2%)	22 (9.2%)	0.69 (0.45–1.08)	0.101
ACEi or ARB	138 (20.4%)	95 (21.7%)	43 (18.0%)	0.87 (0.62–1.21)	0.407
Non-steroidal anti-inflammatory drugs	23 (3.4%)	16 (3.7%)	7 (2.9%)	0.79 (0.37–1.68)	0.542
Clinical presentation:					
Asymptomatic	90 (13.3%)	82 (18.8%)	8 (3.4%)	0.19 (0.09–0.39)	<0.001
Fever	431 (63.8%)	253 (57.9%)	178 (74.5%)	1.86 (1.36–2.54)	<0.001
Respiratory symptoms ^a^	373 (55.2%)	218 (49.9%)	155 (64.9%)	1.55 (1.18–2.03)	0.002
Dyspnea	257 (38.0%)	128 (29.3%)	129 (54.0%)	2.34 (1.79–3.06)	<0.001
Gastrointestinal symptoms ^b^	97 (14.4%)	65 (14.9%)	32 (13.4%)	0.87 (0.59–1.26)	0.451
Pneumonia	435 (64.4%)	237 (54.2%)	198 (82.9%)	3.96 (2.70–5.82)	<0.001
Lymphopenia	481 (71.2%)	281 (64.3%)	200 (83.7%)	3.56 (2.27–5.59)	<0.001
Hospitalization	515 (76.2%)	291 (66.6%)	224 (93.7%)	7.38 (4.12–13.20)	<0.001
Hospitalization days	11 (IQR: 7–17)	14 (IQR: 9.5–21.5)	8 (IQR: 4–12)	0.60 (0.53–0.68) ^e^	<0.001
Admission to ICU	9 (1.3%)	1 (0.2%)	8 (3.4%)	2.29 (1.13–4.64)	0.022
Mechanical ventilation	50 (7.4%)	5 (1.1%)	45 (18.8%)	3.83 (2.72–5.39)	<0.001

^a^ Included cough, rhinorrhea and sputum production. ^b^ Included nausea, vomiting and diarrhea. ^c^ HR estimated for 10-year age increase. ^d^ HR estimated for one year of haemodialysis. ^e^ HR estimated for 5 days of hospitalization.

**Table 4 jcm-10-03022-t004:** Multivariate Cox regression of risk factors for mortality in haemodialysis elderly patients with COVID-19. The hazard ratio (HR) for mortality and its 95% confidence interval are shown for each factor included in the regression model.

Variable	Model 1			Model 2			Model 3		
	HR	95% CI	*p*-Value	HR	95% CI	*p*-Value	HR	95% CI	*p*-Value
Age	1.59 ^b^	1.31–1.93 ^b^	<0.001	1.87 ^b^	1.48–2.37 ^b^	<0.001	1.48 ^b^	1.21–1.82 ^b^	<0.001
Male sex	1.21	0.90–1.63	0.210	1.19	0.85–1.66	0.324	1.16	0.84–1.60	0.369
Clinical presentation:									
Asymptomatic	0.96	0.39–2.35	0.931	1.47	0.53–4.07	0.461	1.31	0.53–3.26	0.562
Fever	1.16	0.81–1.67	0.419	1.36	0.88–2.12	0.165	1.26	0.84–1.90	0.263
Respiratory symptoms ^a^	0.97	0.70–1.34	0.853	0.97	0.67–1.42	0.881	0.92	0.65–1.30	0.619
Dyspnea	1.51	1.11–2.04	0.008	1.57	1.10–2.25	0.013	1.63	1.17–2.26	0.004
Pneumonia	1.74	1.10–2.73	0.017	1.79	1.03–3.11	0.040	2.32	1.38–3.90	0.002
Lymphopenia	1.44	0.87–2.38	0.155	1.58	0.84–2.97	0.152	1.40	0.81–2.43	0.230
Hospitalization	4.00	1.83–8.70	<0.001	4.43	1.77–11.11	0.001	6.69	2.59–17.30	<0.001
Primary kidney disease:									
Diabetic kidney disease				1.57	0.98–2.53	0.063			
Primary glomerular disease				1.07	0.58–2.00	0.826			
Interstitial nephropathy				0.56	0.23–1.37	0.205			
Polycystic kidney disease				1.57	0.67–3.66	0.295			
Nephrosclerosis				1.03	0.61–1.74	0.922			
Systemic disease				2.50	1.06–5.94	0.035			
Treatments received for infection:									
Lopinavir/Ritonavir							0.94	0.67–1.32	0.711
Hydroxychloroquine							0.71	0.45–1.10	0.127
Interferon beta							1.56	0.82–2.98	0.174
Tocilizumab							0.87	0.42–1.82	0.712
Glucocorticoids							0.68	0.48–0.96	0.027
Azithromycin							0.85	0.62–1.18	0.336

^a^ Included cough, rhinorrhea and sputum production. ^b^ HR estimated for 10-year age increase.

## Data Availability

Data from the Spanish general population affected by COVID-19 and showed in this manuscript was obtained from the Spanish National Epidemiological Surveillance Network: https://cnecovid.isciii.es/ (accessed on 1 March 2021).

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
