# Peer review of "The Impact of Age on Mortality in Chronic Haemodialysis Population with COVID-19"

_jcm, 2021, doi:10.3390/jcm10143022_

Round 1
Reviewer 1 Report
The points raised in the previous review have been addressed.
Author Response
We thank the reviewer for his/her comments regarding the manuscript.
Reviewer 2 Report
In this retrospective observational study, Dr. Vergara and colleagues explored the impact of older age on mortality among haemodialysis (HD) patients with COVID-19. As the results, they reported that the all-cause mortality was higher in elderly with HD and the number was higher than the general population. Overall, this could be an insightful study, however several problems are detected. Additionally, I have some questions / suggestions for the authors:
- The writing must be improved. The English contains too many flaws so that they impair the readability and clarity of the manuscript. I will expose some of them below but it would be better to seek help from a scientific writer or professional editing service.
- What did the authors do to rule out the potential influence of lower immune system in elderly? I can see that the presentation of pneumonia and lymphopenia was higher in > 65 y. This would be associated with the severity of COVID-19 and mortality.
- What was the rationale of choosing all-cause mortality? We know that in general, the risk of death in old people are higher than young people.
- When comparing with general populations, how similar are those two populations? How similar was the COVID-19 severity? How similar was the baseline characteristic? Hospitalization rate? etc. Otherwise, the comparison is not apple-to-apple. Please also add a table comparing the baseline characteristics of these groups (general pop vs. HD pop)
- Regarding the statement in lines 52-53, I think if the authors want to study the contribution of CKD as a risk factor of COVID-19 mortality, it was not captured in this study design. The authors could compare COVID-19 patients with / without HD (comparing with general population is interesting but because the population is not screened, it is difficult to interpret the findings). So, please revise the introduction to be more focus with the aim of this study.
- Regarding the statement in lines 53-56, what are the relevance of those sentences about the number of HD patients stratified by age? was this studied in this manuscript? In general, I am confused by the way the authors build the story. I can sense that the authors want to know the effect of CKD and hemodialysis in COVID-19 mortality but unfortunately, it is not well-explored in this study. We have to keep in mind that all of the subjects are under hemodialysis.
- What was the "radiological image" used to diagnose the pneumonia? Rontgen/X-ray? CT? MRI?
- How did the authors assess the severity of COVID-19? Please explain
- How could the authors explain this observation "Nevertheless, once admitted, longer hospital stays were associated with a better prognosis". In almost all diseases, longer hospital stay is always associated with more complications and mortality. Please explain.
- In general, I am afraid that everything we see in the manuscript is not about older or younger but more about severe vs. non-severe COVID-19. Please comment on this concern.
- Section 2.3 can be merged in section 2.1. no need to separate them.
- Line 127: What is non-parametric distribution? I never heard it.
- I suggest the authors to incorporate/discuss the information in this preprint. It highlights the importance of severity and showed that corticosteroids are beneficial in severe state only (https://www.preprints.org/manuscript/202104.0022/v1)
- I am curious to see another table similar to table 2 but for HD patients age <65 y. Please add and compare the hazard ratio with the data in table 2.
- Figure 2B is quite similar to figure 2A. I am not sure what is it for? If the authors want to show the effect of age in various ranges, we will need more ranges there. The authors could use 10 years interval (0-10, 11-20 etc.) to show it better.
- Please add the figures and tables in the text, not the end of the manuscript per MDPI regulation (check the guide for authors).
- Line 24: remove "old"
- Lines 24-25: remove "haemodialysis patients"
- Line 29: should be "hospital admission"
- Line 30: should be "elderly population undergoing hemodialysis"
- Line 38: "...as the cause..."
- Line43: should be "health crisis". By the way, "sanitary crisis" means another thing.
- Line 46: should be "...have been attributed to worse outcomes ..."
- Line 51: should be "of" not "for"
- Line 64: remove "disease"
- Lines 76-77 has to be rephrased. I don't understand the meaning of the sentence.
- Line 78: I don't think "filling the gap" is the right words for this. Maybe "fill in the box?"
- What is the meaning of this sentence in lines 101-102: Physicians considered in most pa- 101 tients the date of discharge as the end of follow-up.
- Line 180: "for" should be "due to"
Author Response
Response to reviewer and editor:
We would like to thank the editor and the reviewer for their comments and suggestions. We have revised the manuscript entitled “The impact of age on mortality in chronic haemodialysis population with COVID-19” and we believe that the revised version of the manuscript has improved substantially. Following, there is a detailed description and answer to the comments and changes proposed by the reviewer.
Reviewer 2:
Comment 1, The writing must be improved. The English contains too many flaws so that they impair the readability and clarity of the manuscript. I will expose some of them below, but it would be better to seek help from a scientific writer or professional editing service.
Response: We thank the reviewer for his/her comments. The English language has been thoroughly revised by a native English speaker, and we think readability of the manuscript has considerably improved.
Comment 2, What did the authors do to rule out the potential influence of lower immune system in elderly? I can see that the presentation of pneumonia and lymphopenia was higher in > 65 years. This would be associated with the severity of COVID-19 and mortality.
Response: We thank the reviewer’s comments. Due to the characteristics of the Registry, information recorded for each patient was limited. Maybe elderly patients show a more severe immunosuppression state manifested by an increased frequency of lymphopenia, or perhaps lymphocyte count is more affected by the infection in this population. A considerable percentage (64%) of non-elderly group also showed lymphopenia. In addition, in other papers it has been associated with mortality (PMID: 33945642 and PMID: 32437770). Thus it is expected to be higher in our older, hemodialysis with COVID-19, a population with high risk of mortality.
A comment has been added in the discussion section to highlight this possible immunosupresion state in elderly population (Lines 454-356).
Comment 3, What was the rationale of choosing all-cause mortality? We know that in general, the risk of death in old people are higher than young people.
Response: We appreciate the reviewer’s comments. Only a limited number of variables were recorded for each patient to ensure adequate completion of the Registry and the participation of different centres during the pandemic (Sánchez-Álvarez et al.; PMID 32389518). Trying to ascertain whether the patient died from COVID-19 or from the exacerbation of previous conditions in an infectious setting could be cumbersome and was not collected in the national registry data. We are sorry but only all-cause mortality was recorded in the Registry. Considering that COVID-19 is an acute illness, this choice should not bias the results. Looking at the Kaplan-Meier survival curves, it is observed that almost all patients died in the first 50 days after diagnosis, reflecting that mortality was due to COVID-19. Nevertheless, a comment has been added to the limitations section (Lines 522-526).
Comment 4, When comparing with general populations, how similar are those two populations? How similar was the COVID-19 severity? How similar was the baseline characteristic? Hospitalization rate? etc. Otherwise, the comparison is not apple-to-apple. Please also add a table comparing the baseline characteristics of these groups (general pop vs. HD pop).
Response: We thank the reviewer for taking out this point. In Spain and European registries (Kramer A et al.; PMID 29423210), the haemodialysis population is normaly older. Therefore, when comparing the characteristics of our haemodialysis population with the general population, patients in dialysis are older and male sex is more frequent. These findings explain in part the increased mortality of our population. However, after adjusting mortality to age, an increased risk of death was observed in every age range for heamodialysis population. Therefore, patients in heamodialysis will probably suffer a more severe infection with an increased risk of death when compared to the general population.
Data from the National Epidemiological Surveillance Network collected a reduced number of variables for each COVID-19 positive case that does not allow us the bseline caractheristics of the patients. It is important to mention that the majority of hemodialysis patients are male, and studies have also been demonstrated a high risk after COVID-19 infection in male as compared to female. In the new version of the manuscript a new table 2 and some comments have been added regarding this point (Lines 97-103 and Lines 423-426).
Comment 5, Regarding the statement in lines 52-53, I think if the authors want to study the contribution of CKD as a risk factor of COVID-19 mortality, it was not captured in this study design. The authors could compare COVID-19 patients with / without HD (comparing with general population is interesting but because the population is not screened, it is difficult to interpret the findings). So, please revise the introduction to be more focus with the aim of this study.
Response: We appreciate the reviewer’s comments. We thank for taking out this point because the objectives written in the first version of the manuscript did not focus on the main objectives of the present manuscript. The last paragraph in the Introduction section has been modified to describe the current study aims properly (Lines 66-71).
Comment 6, Regarding the statement in lines 53-56, what are the relevance of those sentences about the number of HD patients stratified by age? was this studied in this manuscript? In general, I am confused by the way the authors build the story. I can sense that the authors want to know the effect of CKD and hemodialysis in COVID-19 mortality but unfortunately, it is not well-explored in this study. We have to keep in mind that all of the subjects are under hemodialysis.
Response: We thank the reviewer for taking out this important point. We agree that the aims at the Introduction section were not clearly explained and they have been modified for clarity in the new version of the manuscript. In the present study, we aimed to assess the impact of COVID-19 in mortality in older patients on hemodialysis. For that reason, we focused all our analysis in elderly haemodialysis population.
Comment 7, What was the “radiological image” used to diagnose the pneumonia? Rontgen/X-ray? CT? MRI?
Response: We appreciate the reviewer comments. Radiological images were obtained by different techniques depending on the centre where patients were assisted. The most usual technique was chest X-ray. However, in some hospitals computed tomography or other procedures have also been used. A comment clarifying this point has been added in the new version of the manuscript (Lines 166-168).
Comment 8, How did the authors assess the severity of COVID-19? Please explain
Response: We thank the reviewer for his/her comments. In the present Registry, only a limited number of variables were recorded. The severity of COVID-19 was assessed by signs, symptoms and progression of the disease. Variables such as dyspnea, pneumonia or the need for mechanic ventilation were used to assess the severity of COVID-19 in our dialysis population. This is now mentioned in the new version of the manuscript (see lines 171-174).
Comment 9, How could the authors explain this observation “Nevertheless, once admitted, longer hospital stays were associated with a better prognosis”. In almost all diseases, longer hospital stay is always associated with more complications and mortality. Please explain.
Response: We appreciate the reviewer’s comments. In ESKD patients, COVID-19 has high mortality in the first week after admission to hospital (Valeri et al.; PMID 32467113). Although they had longer hospital stays, patients who survived the first week of hospitalization had a greater probability of being discharged. This has been now discussed in the new version of the manuscript (see lines 370-373).
Comment 10, In general, I am afraid that everything we see in the manuscript is not about older or younger but more about severe vs. non-severe COVID-19. Please comment on this concern.
Response: We thank the reviewer for his/her kind comments. Although it was expected, our hypothesis confirms that the combination of older age and hemodialysis has a significant impact on COVID-19 mortality. Moreover, we also showed that elderly patients affected by COVID-19 on hemodialysis have at least a 2-fold increase in mortality compared to elderly patients from the general population. This finding is probably related to a more severe disease in older haemodialysis patients, as they depicted a higher prevalence of dyspnea at presentation or pneumonia.
Comment 11, Section 2.3 can be merged in section 2.1. no need to separate them.
Response: We thank the reviewer for his/her comments. Changes have been made in the new version of the manuscript and these two sections have been merged.
Comment 12, Line 127: What is non-parametric distribution? I never heard it.
Response: We thank the reviewer for his/her comments. We were mistaken on this point. It is non-normal distribution and non-parametric tests. This has been now corrected in the new version of the manuscript. (line 181)
Comment 11, I suggest the authors to incorporate/discuss the information in this preprint. It highlights the importance of severity and showed that corticosteroids are beneficial in severe state only (https://www.preprints.org/manuscript/202104.0022/v1)
Response: We thank the reviewer for his/her kind comments. The reference has been added to the new version of the manuscript (see REF 24).
Comment 12, I am curious to see another table similar to table 2 but for HD patients age <65 y. Please add and compare the hazard ratio with the data in table 2.
Response: We thank the reviewer for the comments. A supplementary table 1 has been added with the requested data. The table has not been included in the manuscript, because our study is focused in elderly haemodialysis patients.
Comment 13, Figure 2B is quite similar to figure 2A. I am not sure what is it for? If the authors want to show the effect of age in various ranges, we will need more ranges there. The authors could use 10 years interval (0-10, 11-20 etc.) to show it better.
Response: We appreciate the reviewer’s comments. Figure 2B has been modified in the new version of the manuscript, and it includes age ranges with 20 years interval.
Comment 14, Please add the figures and tables in the text, not the end of the manuscript per MDPI regulation (check the guide for authors).
Response: We thank the reviewer for his/her comments. Changes have been made in the new version of the manuscript according to this point and MDPI regulation.
Comment 15, Line 24: remove “old”
Response: We thank the reviewer for his/her comments. This has been modified in the new version of the manuscript.
Comment 16, Lines 24-25: remove “haemodialysis patients”
Response: We thank the reviewer for his/her comments. Changes have been made in the new version of the manuscript.
Comment 17, Line 29: should be “hospital admission”
Response: We thank the reviewer for his/her comments. Changes have been made in the new version of the manuscript.
Comment 18, Line 30: should be “elderly population undergoing hemodialysis”
Response: We thank the reviewer for his/her comments. Changes have been made in the new version of the manuscript.
Comment 19, Line 38: “...as the cause...”
Response: We thank the reviewer for his/her comments. Changes have been made in the new version of the manuscript.
Comment 20, Line 43: should be “health crisis”. By the way, “sanitary crisis” means another thing.
Response: We thank the reviewer for his/her comments. Changes have been made in the new version of the manuscript.
Comment 21, Line 46: should be “...have been attributed to worse outcomes ...”
Response: We thank the reviewer for his/her comments. Changes have been made in the new version of the manuscript.
Comment 22, Line 51: should be “of” not “for”
Response: We thank the reviewer for his/her comments. Changes have been made in the new version of the manuscript.
Comment 23, Line 64: remove “disease”
Response: We thank the reviewer for his/her comments. Changes have been made in the new version of the manuscript.
Comment 24, Lines 76-77 has to be rephrased. I don’t understand the meaning of the sentence.
Response: We appreciate the reviewer comments. The sentence has been rewritten it’s understanding easier.
Comment 25, Line 78: I don’t think “filling the gap” is the right words for this. Maybe “fill in the box?”
Response: We thank the reviewer for his/her comments. Changes have been made in the new version of the manuscript.
Comment 26, What is the meaning of this sentence in lines 101-102: Physicians considered in most patients the date of discharge as the end of follow-up.
Response: We thank reviewer’s comments. The sentence has been rewritten in the new version of the manuscript.
Comment 27, Line 180: “for” should be “due to”
Response: We thank the reviewer for his/her comments. Changes have been made in the new version of the manuscript.
Round 2
Reviewer 2 Report
Thank you for addressing my comments.
Author Response
See attached document.

This manuscript is a resubmission of an earlier submission. The following is a list of the peer review reports and author responses from that submission.
Round 1
Reviewer 1 Report
I feel the data presented in your paper has been (at least partly) presented in other papers written from same registry:
Sánchez-Álvarez JE, Fontán MP, Martín CJ, Pelícano MB, Reina CJC, Prieto ÁMS, Melilli E, Barrios MC, Heras MM, Pino MDDPY. Status of SARS-CoV-2 infection in patients on renal replacement therapy. Report of the COVID-19 Registry of the Spanish Society of Nephrology (SEN). Nefrología (English Edition). 2020 May-June;40(3):272–8. doi: 10.1016/j.nefroe.2020.04.002. Epub 2020 Apr 27. PMCID: PMC7183935.
Jager KJ, Kramer A, Chesnaye NC, Couchoud C, Sánchez-Álvarez JE, Garneata L, Collart F, Hemmelder MH, Ambühl P, Kerschbaum J, Legeai C, Del Pino Y Pino MD, Mircescu G, Mazzoleni L, Hoekstra T, Winzeler R, Mayer G, Stel VS, Wanner C, Zoccali C, Massy ZA. Results from the ERA-EDTA Registry indicate a high mortality due to COVID-19 in dialysis patients and kidney transplant recipients across Europe. Kidney Int. 2020 Dec;98(6):1540-1548. doi: 10.1016/j.kint.2020.09.006. Epub 2020 Oct 15. PMID: 32979369; PMCID: PMC7560263.
The other concern I have is that the novelty of your findings: the impact of age on COVID-related mortality is low, therefore I recommend additional data analysis, as the dataset is impressive and considering asking an important question before looking into the data.
Reviewer 2 Report
The aim of this study is to investigate the impact of older age on mortality among hemodialysis patients with COVID-19.
There are previous reports highlighting the high mortality in old patients on hemodialysis diagnosed with COVID-19. However, the authors analyzed in a large number of patients the differences in mortality between the elderly hemodialysis population and the elderly general population, mortality risk, and the specific treatments for COVID-19 administered to the elderly population on hemodialysis. These analyses extend the scientific interest of this manuscript
Although the experimental design is adequate, there are minor concerns in the presentation and the interpretation of the results.
Following are some specific concerns about the manuscript:
- Line 55. KRT, abbreviations should be defined at first mention in the text.
- The high mortality in the elderly hemodialysis patients probably was also affected because these patients did not receive more invasive or intensive care therapies compared to non-elderly hemodialysis population. A comment about this difference could be included in the discussion.
